# Gut-on-a-Chip Research for Drug Development: Implications of Chip Design on Preclinical Oral Bioavailability or Intestinal Disease Studies

**DOI:** 10.3390/biomimetics8020226

**Published:** 2023-05-28

**Authors:** Joanne M. Donkers, Jamie I. van der Vaart, Evita van de Steeg

**Affiliations:** 1Department of Metabolic Health Research, TNO, Sylviusweg 71, 2333 BE Leiden, The Netherlands; j.i.van_der_vaart@lumc.nl (J.I.v.d.V.); evita.vandesteeg@tno.nl (E.v.d.S.); 2Division of Endocrinology, Department of Medicine, Leiden University Medical Center, Albinusdreef 2, 2333 ZA Leiden, The Netherlands; 3Einthoven Laboratory for Experimental Vascular Medicine, Leiden University Medical Center, Albinusdreef 2, 2333 ZA Leiden, The Netherlands

**Keywords:** organ-on-a-chip, gut-on-a-chip, intestine, in vitro, ex vivo, ADME, oral bioavailability, drug development, microbiome, IBD

## Abstract

The gut plays a key role in drug absorption and metabolism of orally ingested drugs. Additionally, the characterization of intestinal disease processes is increasingly gaining more attention, as gut health is an important contributor to our overall health. The most recent innovation to study intestinal processes in vitro is the development of gut-on-a-chip (GOC) systems. Compared to conventional in vitro models, they offer more translational value, and many different GOC models have been presented over the past years. Herein, we reflect on the almost unlimited choices in designing and selecting a GOC for preclinical drug (or food) development research. Four components that largely influence the GOC design are highlighted, namely (1) the biological research questions, (2) chip fabrication and materials, (3) tissue engineering, and (4) the environmental and biochemical cues to add or measure in the GOC. Examples of GOC studies in the two major areas of preclinical intestinal research are presented: (1) intestinal absorption and metabolism to study the oral bioavailability of compounds, and (2) treatment-orientated research for intestinal diseases. The last section of this review presents an outlook on the limitations to overcome in order to accelerate preclinical GOC research.

## 1. Introduction

The mainly used and most convenient way of systemic drug delivery is the oral route, assigning a key role to the gut for accommodating the uptake, transport, and in some cases, the metabolism of drugs. The gut is therefore seen as one of the main target organs for drug metabolism and pharmacokinetic (DMPK) research, and together with the liver and kidneys, the intestines influence the absorption, distribution, metabolism, and elimination (ADME) of drug candidates. Therefore, it is not surprising that the use of intestine-specific in vitro research models is widely adopted in preclinical drug development to investigate individual processes, mechanisms of action and the prediction of the oral bioavailability of novel drug entities in the human body.

Primarily known for its digestive and nutrient absorptive functions, the gut serves as an exquisite barrier between the external environment and the inside body and is capable of letting beneficial compounds pass, e.g., nutrients and drugs, but protecting our body against harmful substances and pathogens [1]. Equipped with multiple specialized cell types for not only nutrient or drug absorption but also hormone secretion and mucus production, the intestinal epithelial barrier has a complex 3D-architecture of villi and microvilli that differs significantly from the upper to lower gastrointestinal (GI) tract [2]. Furthermore, resident immune cells within the gut tissue and the large microbial community inhabiting the gut wall further increase the complexity of this organ [3].

Recapitulating the complete gut physiology and architecture in vitro has therefore proven to be difficult, and most conventional models reflect at best only part of the intestinal characteristics, e.g., the epithelial lining. Traditional 2D cell culture of immortalized epithelial cell lines such as Caco-2, HT-29 and T84 in a Transwell setup is still seen as the golden standard method for in vitro drug absorption and metabolism studies [4,5,6], although highly limited by its monoculture origin and incorrect reflection of the in vivo expression of transporters and metabolizing enzymes [7]. For example, the expression of carboxylesterase 1 (CES1) in these cells is higher compared to carboxylesterase 2 (CES2), thereby following a liver-specific expression pattern whereas the opposite is true for the intestines and thus incorrectly predicting metabolism of certain drugs, such as mycophenolate mofetil [8]. Since 2009, 3D culturing of so-called organoids derived from primary intestinal crypts became popular as they better reflect the intestinal cell type diversity and architecture [9] and if cultured in 2D monolayers in Transwell setups, they can be used to study intestinal drug absorption and metabolism [10,11]. These organoid-derived monolayers show higher expression and functionality of major pharmacokinetic enzymes and transporters compared to Caco-2 monolayers, at levels close to the in vivo human intestine [12]. Alternative to these cell-based in vitro research models, ex vivo tissue-based models such as the Ussing chamber [13,14,15], the everted sac model [16] and the InTESTine™ system [17,18] or animal models are also used to study drug-related processes in the gut. Although these models reflect the true intestinal physiology, they are limited in throughput and lifespan (ex vivo) or ethics and translatability to humans (animal models). Clearly, there is room for improvement, and in that light, the rapid emerging novel microfluidic organ-on-a-chip (OOC) technology might be considered as the next step for in vitro research to study the fate of drug candidates in the gut. The OOC contains continuously perfused chambers where cells, organoids or tissue can be cultured to study processes in an improved physiological setting on the tissue and organ levels [19]. The OOC system can recreate key features of the complex organ microenvironment such as shear stress and 3D architecture and could even be combined in a multi-OOC [20,21].

Here, we will focus on the intestine-on-a-chip, or more commonly named the gut-on-a-chip (GOC). There are many different GOC models, each with their own advantages and disadvantages. This review will summarize the important considerations when selecting and designing a GOC with potential preclinical applications in drug development research. Step-by-step, we describe the choices that have to be made for, among others, the chip material and fabrication process, cell source and tissue engineering, environmental factors and read-out possibilities in the chip. Examples of GOC applications for, in particular, preclinical oral bioavailability research are described next, as well as examples of using GOCs to study intestinal disease.

## 2. The Four Components Influencing GOC Design

Currently, a ‘standard’ GOC does not exist that on one hand hampers the comparability between studies but on the other hand also allows a huge amount of freedom to design your chip to your specific needs. In GOC engineering, the four most important components that should be taken into consideration are (1) the biological research question, (2) chip material, (3) tissue engineering, and (4) the readout parameters (Figure 1). For example, a simple GOC model can be used to study absorption, whereas a more complex and sophisticated model is needed to study disease processes. Furthermore, the unique properties of each material used to fabricate the chips also reflect its application, e.g., the elasticity of the plastic chip material determines part of the architecture of the cells or tissue on the GOC. Furthermore, the cell source that is used determines not only the effectiveness of answering the research question, but also the readout parameters. The next subsections elaborate on the possibilities of each of the four components influencing GOC engineering and discuss their advantages and disadvantages.

### 2.1. The Biological Research Question Determines the Type of GOC to Use

The first step in any study is to formulate the biological research question, which sets the criteria for the materials, biomaterial and microenvironment that should be included or applied. By using a stepwise approach in the selection process of the application, endpoints, throughput and experimental timeline, the required components minimally needed to answer a biological question result in the design of a minimal viable, but not necessarily the most optimal, GOC model. For example, to investigate toxicity of a novel drug entity, a chip made of impermeable material such as polymethyl methacrylate (PMMA), combined with a 2D monolayer of Caco-2 cells would be the bare minimum to build the GOC model [22,23]. In general, simpler models with one cell type, a simple extracellular matrix (ECM), and few readout parameters are used for short high throughput experiments, such as drug toxicity and absorption tests. Kimura et al. (2008) described one of the first GOC models used for compound permeability screening, culturing Caco-2 cells in a polydimethylsiloxane (PDMS) chip showing transport of rhodamine 123 [24]. Additional model complexity with ex vivo tissue or multiple cell types and complex ECM could be added to better understand the mechanism of absorption or study long-term drug effects on, for example, gut health [23,25,26]. More complex GOC models often have a lower throughput, require extra costs and might increase experimental variability as more variables are added to the system. Thus, the most optimal GOC design will differ per research question. Nevertheless, many research groups have established their own GOC model, reusing it with small alterations for new research questions without considering new or alternative materials or components for their GOC model that can impact study results.

### 2.2. Fabrication and Materials for the Individual Chip Components

When designing a GOC, or choosing from the selection of commercially available GOCs, the so-called spatiotemporal parameters such as the desired culture period and size and shape of the culture chamber should be considered first. For example, when cells are cultured for a longer period of time in combination with mechanical and biochemical cues, cells are able to secrete their own ECM, remodel biomaterial, and differentiate faster into different intestinal cell types (goblet cells, enterocytes, Paneth cells, and neuroendocrine cells) [27,28,29,30]. However, extensive culture periods also increase the risk on de-differentiation, contamination and the chance on hypoxic or necrotic centers [31,32]. Furthermore, designs optimized for long culture periods are probably less useful for tissue-based systems as they show a maximum culture period of only 3 days [25,33]. An additional important factor for cells is that they can be grown in 2D or 3D structures either on a flat surface or on a structured surface with for example a villi-like architecture [34,35]. Additionally, the required flow and sample size may also impact the size and shape of the culture chamber. Once the design of the chip is ready, a choice needs to be made for the fabrication technique and materials for the three main components of the GOC device: chip slide, membrane, and ECM.

#### 2.2.1. Fabrication Techniques

Photolithography, oxidation, e-beam evaporation, wet etching, sputtering, injection molding, micro molding, electrospinning, laser ablation, computer numerical control machining, hot embossing, CO_2_ laser engraving and 3D printing are all methods that are applied to engineer GOC devices [36]. To discuss all techniques in detail is beyond the scope of the review, but here, we describe the three main fabrication techniques used for GOC: photolithography, 3D (bio)printing and electrospinning.

Photolithography involves applying UV light onto UV-sensitive material to create a pattern [19]. It is the most commonly used technique for fabricating GOC and other OOC devices. In practice, photolithography is often combined with micro molding, to generate a soft and flexible patterned structure. For example, PDMS can be micro molded into a multi-layered structure on a mold made with photolithography [27]. This technique is especially helpful to direct certain cell migration patterns, thus controlling the spatial architecture.

With the possibility to easily modify designs and select different printing materials, 3D printing holds a great promise for the OOC field. With stereolithography (SLA) 3D printing, more complex mechanical mechanisms and micro structured molds can be created in the chip devices compared to conventional 3D printing [25,37]. Often, a 3D polymer structure is made from a liquid photopolymer that is hardened using a UV laser. Examples of materials used for 3D printed GOC models are acrylonitrile butadiene styrene (ABS) [38], hydrogels [39] and a resin used in dental applications [25]. In addition, 3D printing can be used via an indirect approach by printing molds and casting the final materials in the mold cavity [40]. Apart from using 3D printing to generate the chip material, biological materials could also be printed onto GOC devices. Although not yet fully developed, this laser-based technique of printing a cell substrate onto a biological gel holds great potential [37]. It is for example highly suitable to generate vascular networks on a chip system, but so far, it is less common for printing multi-layered organs such as intestinal tissue. Nevertheless, it has been incorporated in a few GOC models to create a villi structure consisting of collagen and Caco-2 cells with a 90% cell viability and biomarker expression of differentiated epithelium [41].

Unlike the photo-based techniques, electrospinning makes use of high electrostatic forces to generate polymeric fibers, which are then injected into a mold [19,42]. Fiber orientation and thickness can be closely controlled on a nanoscale level, and it is especially suitable to generate the membranes separating the intestinal and endothelial cells. This technique is favorable for cell migration and infiltration studies, but often this is no special need for GOC studies.

#### 2.2.2. Chip Slide Material

The major considerations when selecting the chip slide material include cell toxicity, gas permeability, transparency, costs, and adhesion strength [43,44]. Early GOCs were often made from glass, a material to which cells easily attach. Glass has the advantage of being transparent, non-toxic to cells, compatible with many solvents and it does not absorb other molecules; however, it is relatively expensive, prone to breaking and difficult to sculp into specific structures. Despite the developments of Liquid glass [45] and laser ablation techniques [46] to generate 3D glass structures, budget and time-requirement reasons initiated the shift of glass to polymers during the late 1990s [47].

To date, PDMS is the most commonly used chip material for GOCs. It is a synthetic polymeric thermoplastic elastomer based on silicone that can be commercially purchased as two separate fluids, which, once combined, crosslink and can be poured into a mold to form complex 3D structures [19,47]. The benefits of PDMS include that it is non-toxic to cells; it is gas-permeable, thus allowing oxygen to flow to the cells; and it is optically transparent, enabling cell microscopy. One of the major disadvantages of PDMS is its strong hydrophobicity that hampers cell attachment and leads to undesired absorption of biomolecules, resulting in decreased effects of these molecules and incorrect predictions of oral bioavailability in intestinal permeability studies [48,49]. Furthermore, the shrinkage of PDMS during the fabrication process hinders standardization and the speed-up of chip manufacturing, and its porous and gas-permeable nature complicates studies using oxygen gradients (e.g., one side anaerobic conditions) and allows water to evaporate, thereby affecting compound concentrations [50]. Although the use of coatings has been suggested, e.g., a layer of parylene [51], paraffin wax [52] or LipoCoat Cellbinder [53] to reduce undesired absorption, this is not widely adopted.

As a first alternative option, other thermoplastic elastomers with properties similar to PDMS such as thermoset polyester (TPE), polyurethane methacrylate (PUMA), Norland Adhesive 81 (NAO81), styrene-(ethylene/butylene)-styrene (SEBS) and tetrafluorethylene-propylene (FEPM) might be considered [54,55,56]. Furthermore, also non-elastic thermoplastics could be used such as PMMA, polycarbonate (PC), polystyrene (PS), polyvinyl chloride (PVC), polyimide (PI), and the family of cyclic olefin polymers (COP), which all have the major advantage of being less expensive, more rigid, and resistant to permeation of chemicals [47], as well as the materials described in the 3D printing paragraph. Thus, there are plenty and possible more suitable alternatives to PDMS, but due to its low costs and ease-of-use, PDMS will probably remain the most used material for GOC devices in the near future.

#### 2.2.3. Membrane Material

An important feature of GOC is the porous membrane separating the apical and basolateral sides, often coated with hydrogels to simulate the ECM. Similarly, to the chip slide, membranes are regularly made of thermoplastic materials, such as PC, polytetrafluorethylene (PTFE, Teflon), polyethylene terephthalate (PETE) and polyester (PE) [57]. Currently, PC coated with collagen is mostly used in GOC devices. However, PC is prone to erosion and can release the toxic bisphenol A [58]. Transparency of the membranes is a challenge when live cell imaging under a microscope is desired. PC, PTFE and PETE are all auto-fluorescent and thus less suitable for microscopy studies. As for PDMS, more research is needed to find low-cost alternatives with similar or better properties. Moreover, with sustainability in mind, research might also focus on the development of non-plastic alternatives that are environmentally friendly and biodegradable [59].

#### 2.2.4. Extracellular Matrix (ECM)

The main physiological function of the ECM is to provide support and a physical scaffold for cells [60,61]. In addition, the ECM regulates the microenvironment around cells by providing chemical signals for proliferation, differentiation, morphogenesis, and homeostasis. The composition of ECM is complex and differs highly between tissues and organisms, but in general consists of high molecular weight proteins such as collagen and fibronectin and branched glycosaminoglycan structures such as heparin sulphate and chondroitin sulphate.

In GOC, the ECM is usually a hydrogel. Hydrogels are 3D networks based on small molecule or polymer building blocks capable of holding large amounts of water [62]. The most important characteristics of hydrogels are their elasticity, permeability, porous but rigid enough structure to support cell cultures. They come from various natural sources but can also be engineered synthetically [62,63].

Natural hydrogels include collagen, gelatin, alginate, fibronectin, and fibrin [60,61]. Collagen is the most frequently used ECM hydrogel in GOC devices. It is the main component of native ECM found in human tissue and provides scaffolding sites, cell binding sites with signals for spreading, growth, differentiation, and proliferation. Gelatin is also regularly used as it has a similar composition to collagen and has the advantage of being less expensive and less antigenic [19,64]. Alginate is derived from algae and seaweed, but despite its low costs and toxicity, it is not very applicable due to its low cell binding sites and high degradability. Fibronectin and fibrin are often used in combination with collagen or gelatin and are especially suitable for scaffolding [61,65]. Although closely mimicking in vivo ECM, natural hydrogels have the downside of high batch-to-batch variation, a need for sterilization and purification, and on the ethical side, the need for animal models when extracting, for example, collagen or gelatin from rat tail or pig skin, respectively [66,67].

In comparison to natural hydrogels, synthetic hydrogels such as polyethylene glycol (PEG) can be modified in their chemical and physical properties, such as the degradation rate, molecular weight, concentration, and crosslinking. Furthermore, the production process can be standardized, resulting in minimal batch-to-batch variation [19]. However, synthetic hydrogels often show a lack of cell binding sites and low surface adhesion molecules, hampering their wide-scale adoption in OOC and other cell-based in vitro platforms. As a result, advances have been made to develop natural and synthetic hybrid hydrogels in order to combine the desirable properties of the natural components with the adaptive ability of synthetic structures.

### 2.3. Tissue Engineering in the GOC

The process of obtaining an artificial functional construct that represents an organ in vitro is also called tissue engineering [68]. More than the material or fabrication process will the choice for a specific cell strain, organoids or tissue explants influence the impact of GOC study results and in vitro to in vivo translatability of the model. Clearly, ex vivo tissue explants represent the in vivo gut tissue the best, as they have the proper architecture and an intact structure with all different gut epithelial cell types represented in the right amounts, as well as containing connective tissue and resident immune cells. Despite having shown good results for sustained tissue viability, intact barrier integrity and drug transport, only a few GOC devices apply ex vivo tissue explants in their model [25,33,69]. The need for a larger amount of fresh donor tissue, difficulty in fixing the explants between the apical and basolateral compartments in a leak-tight manner and limited lifespan of the explants might discourage their use and make researchers choose for a cell-based GOC. However, engineering a gut tissue-on-chip with high complexity involves the incorporation of multiple cell types, of which an example is shown in Figure 2.

#### 2.3.1. Source of the Intestinal Cells

Largely influenced by costs and availability, most GOC models use the traditional immortalized epithelial cell lines Caco-2 and HT29, or a combination of the two. The Caco-2 cell line counts as the most applied cell line for intestinal permeability assays, as these cells spontaneously differentiate into a monolayer with properties typical for the small intestinal enterocytes [70]. Similarly to Caco-2 cells, HT-29 cells were also originally derived from colon carcinoma, and especially, the subline HT29-MTX is representative of the mucus-producing goblet cells [71]. In most cases, they are used in combination with Caco-2 cells as a more accurate model of the in vitro small intestine including the mucus layer [72]. These immortalized cell lines are easy to maintain and can be cultured for many passages without alterations in phenotype, thus supporting the generation of reproducible results. However, a major limitation of these cell lines is that they have an altered transporter and enzyme expression pattern compared to the in vivo gut, and are unable to represent the different intestinal segments (duodenum–jejunum–ileum–colon) with their region-specific functions [7]. Consequently, the limited representation of the in vivo gut using cell line-based GOC may lead to the use of more suitable alternatives.

If cells are still the preferred choice of a researcher to engineer the GOC, biopsy-derived primary cells will more closely resemble the in vivo epithelial layer. However, the extraction of primary cells is difficult and is highly dependent on the availability of donor tissue; the growth of adult cells is challenging as their phenotype may change after a couple of days due to a change in the microenvironment, and they have a short lifespan [19,73]. The use of primary cells for GOC is therefore limited, and interest has shifted mainly towards induced pluripotent stem cells (iPSCs) and organoids as a cell source in the GOC. The 3D organoid cultures are predominantly established from tissue biopsies, subsequently made into a single cell suspension and seeded into the GOC to form villi-like structures. Examples of GOC models using this approach are Intestine Chip [74], Jejunum-intestine Chip [75], Colon Chip [76], Colon-intestine Chip [77] and GuMI [78]. Another route to obtain intestinal organoids is by using iPSCs. These iPSCs are usually obtained from skin, plasma or urine samples and are subsequently reprogrammed into an embryonic-like pluripotent state that allows the differentiation into any desired cell type based on provided growth factors and biochemical cues [79,80,81]. The advantage of non-invasive access in patients opens the window for personalized medicine approaches, and thus, on the study of disease characteristics or therapeutic efficacy at the patient level. However, iPSC-derived organoids have an immature signature, even after differentiation, and differences between intestinal regions are difficult to recapitulate. In contrast, biopsy-derived organoids might be more expensive to culture, but have a mature phenotype and keep their region- and, if applicable, disease-specific characteristics [82]. Interestingly, a recent study showed that the intermediate organoid culture step can be omitted with a direct iPSC culture and differentiation into intestinal tubule-like structures on the OrganoPlate^®^ GOC device [83]. The authors show that their set up is applicable for ADME and intestinal inflammation studies as drug transporter proteins and metabolic enzymes were upregulated, the barrier function was intact, and the tubules showed an inflammatory response upon stimulation.

#### 2.3.2. Co-Culturing with Additional Cells: Endothelial and Immune Cells

While the absorptive properties of the intestine might be represented by using intestinal cells only, increasing number of researchers acknowledge the need for endothelial and immune cells in the GOC as a model of the intestinal vasculature or gut immune system, respectively.

There are several GOC models that incorporate endothelial cells to form an endothelial–epithelial interface with the endothelial cells growing on the basolateral side and the intestinal epithelial growing on the apical side, usually separated by a porous membrane [74,76,84,85,86,87,88,89,90,91,92,93]. As they are growing in different compartments, two medium flows can be maintained containing specific growth factors needed for each cell type, without interfering with cell–cell communication. They were originally added to establish a more physiological model of the tissue–tissue interface in the gut, which might impact the transport of fluids, nutrients and drugs over the epithelial layer [94], and the endothelial cells beneficially influence epithelial cell growth and culture time [74]. Although their function is similar in every GOC, endothelial cell sources differ substantially. Mostly, vascular endothelial cell lines such as Human Umbilical cord Vein Endothelial Cells (HUVECs) and Human Intestinal Microvascular Endothelial Cells (HIMECs) are used, of which the latter is preferred for GOC due to their intestinal background [74].

Immune-related questions in the GOC, for example, to study inflammatory bowel disease (IBD) are usually studied in models that added immune cells to their design. Medium optimization might be necessary for these types of co-cultures to select the most optimal growth and nutritious conditions for all cell types present in the same compartment of the GOC. Generally, the immune cells are added to the basolateral side of the intestinal epithelial cells [88,89,95,96] or to the ECM [97]. Most frequently, peripheral blood mononuclear cells (PBMCs), macrophages, dendritic cells or neutrophils are used. PBMCs are a mixture of innate and adaptive immune cells such as monocytes, dendritic cells and lymphocytes. Examples of studies that applied PBMCs are [89,96], showing that their presence can accelerate the decrease in the barrier integrity and damaging the intestinal epithelium by pathogenic bacteria or bacterial components. Other studies apply only one- or two specific immune cell type(s) such as the macrophage cell line THP-1, dendritic cell line MUTZ-3 [95], primary monocytes differentiated into macrophages and dendritic cells [88] or neutrophils [97]. The activation of the macrophages and/or dendritic cells decreased the barrier integrity of gut tubules [95,97] and attracted neutrophils [97] or dendritic cells [88] that migrated through the ECM to the intestinal epithelial cells. Although these studies show that there is a large variety in different immune cell types to choose from and that this might influence study results, they also endorse that adding an immune component to the GOC is needed when studying complex processes such as the diseased and leaky gut or interaction with (pathogenic) microbiota.

#### 2.3.3. The Gut Microbiome

The largest population of microbial organisms in the human body resides in the intestines and forms a complex ecosystem of bacteria, eukaryotes and viruses [98]. This population is also known as the gut microbiota or gut microbiome, two terms that are used interchangeably. Living on top or in close proximity to the gut epithelial cells, it is clear that the microbiome plays an important role in human health and disease. For example, some bacteria ferment undigestible carbohydrates into short chain fatty acids (SCFA) that serve as an energy source for the intestinal enterocytes [99]. Furthermore, they interact with the gut immune cells in order to maintain the balance between host defense and immune tolerance [98,100]. Dysbiosis, the dysregulation of the host–bacteria homeostasis, is associated with multiple pathological diseases, including metabolic syndrome, obesity, diabetes, autoimmune disorders, allergies, IBD, and colorectal cancer [98,100]. Despite the recognition that characterization of the gut microbiome and its interaction with the host cells is essential, proper analysis of host–microbe interactions remains largely limited to genetic or metagenomic analysis of fecal samples. Hampered by their rapid overgrowth of intestinal cells in static culture systems and the need for strict anaerobic circumstances by many bacterial strains, in vitro host–microbe co-cultures are hardly established in conventional experimental gut models [87]. Flow-enforced formation of a protective mucus layer on top of the intestinal cells and the compartmentalized design of GOCs make them an attractive model to study the host–microbe effects [27]. A distinction can be made between GOCs that have an anaerobic compartment and GOCs that do not have such a compartment and thus can only include probiotic or pathogenic strains that survive in oxygen-rich conditions. Examples of probiotics in a GOC are the co-culture of Caco-2 cells with *Lactobacillus rhamnosus GG* (LGG) [27,85,88] or with a therapeutic combination of 8 probiotic strains called VSL#3 (mixture of *Lactobacillus acidophilus*, *L. plantarum*, *L. paracasei*, *L. delbrueckii subsp. bulgaricus*, *Bifidobacterium breve*, *B. longum*, *B. infantis,* and *Streptococcus thermophilus*) [89,96]. Interestingly, 72 h co-culture of the probiotic mix with Caco-2 cells demonstrated a gene expression profile most similar to in vivo human ileum tissue [89] and a protective effect against barrier disruption and a pro-inflammatory immune response in an IBD GOC model when administered in advance of disease onset [96] or against invasion of the enteroinvasive *Escherichia coli* (EIEC) [89]. A probiotic-related protective effect against pathogen invasion was also demonstrated for *L. rhamnosus* against the fungus *Candida albicans* [88] and partially against extended-spectrum β-lactamase producing *E. coli* (ESBL-EC) [85]. Enhanced mucus production was suggested as an important protective factor, something that was also found in a study that investigated the invasion of the pathogens *Salmonella typhimurium* and *E. coli* (serotype O157:H7) in a GOC [101]. By showing that invasion in a 3D model was different from a 2D model, those and other researchers that investigated *Shigella flexneri* invasion concluded that the GOC microenvironment with 3D architecture and mechanical cues such as shear stress are inevitable to obtain results with a proper translatability to the in vivo situation [101,102]. With an aerobic–anaerobic interface contributing to an even better physiological microenvironment for host-microbe co-cultures, the HMI (Host–Microbiota Interaction) [103], HuMiX (Human–Microbial–crosstalk) [104], AOI Chip (Anoxic-Oxic Interface-on-a-chip) [105], the anaerobic Intestine Chip [87], GuMI (gut microbiome) [78], PMI Chip (Physiodynamic Mucosal Interface-on-a-chip) [106], MihI-oC (Microbiota human-Intestine-axis on Chip) [107] and hypoxic chips [108,109] have demonstrated the survival and metabolic activity of facultative (LGG [103,104,107]) and strict anaerobic (*Bacteroides caccae* [104], *Bifidobacterium adolescentis* [105,108], *B. longum* [107] or *B. bifidum* [109], *Eubacterium hallii* [105], *Bacteroides fragilis* [87] and *Faecalibacterium prausnitzii* [78]) gut commensals without affecting intestinal cell viability. Although most of these models used simple setups with Caco-2 cells and single bacterial strains to mimic the intestine or microbial community, respectively, a more complex model was also established with full human microbiome derived from stool samples co-cultured with primary intestinal cells [87,106]. While the microbial cultures were still physically separated by a porous membrane in the HMI and HuMiX GOCs [103,104], direct cell–cell contact was established in the AOI Chip, anaerobic Intestine Chip, GuMI and the PMI Chip [78,87,105,106]. Apart from the obvious measure to perfuse the apical side of the chip with anaerobic medium, each of these devices used a different approach to keep this side anaerobic in the normal laboratory environment, e.g., by physical separation [103,104], use of oxygen impermeable materials [78,104], increased amount of PDMS on top of the apical chamber [105,108,109], or incorporation of a small anaerobic chamber [87,107]. At the same time, the chip is perfused with aerobic medium, providing the required oxygen to the intestinal cells at the basolateral side, the side to which also other organs can be connected in a multi-organ OoC. The variety of technologies and low number of (follow-up) publications show that there is still much to be gained in this field of GOC research.

### 2.4. Environmental Factors and Read-Out Parameters Important for the GOC Design

In addition to the interaction with neighboring cells as described in the previous section, organ functionality is dependent on a diverse array of environmental factors, including mechanical, electrical, physiochemical, and biochemical signals. Obviously, these factors are absent in vitro and therefore have to be introduced and regulated in GOC models. Additionally, the signals provided by the cells or tissue inside the GOC can function as readout parameters.

#### 2.4.1. Mechanical Cues Important in the GOC

The most important mechanical cues for the gut are shear stress, its peristaltic movement and matrix elasticity [57]. As food is passing through the intestine, the cells experience shear stress and this mechanical signal is converted into a chemical signal that activates cellular processes. Consequently, by simulating shear stress in GOCs, a variety of important cellular and biochemical processes are also activated. Indeed, Caco-2 cells that were exposed to shear stress in a GOC were fully matured in 4–5 days, compared to 21 days in static conditions [27,110]. This was highlighted by the production of villin, a marker for epithelial cell differentiation and brush border formation, and high expression of mucin-2, a biomarker for mucus production [27,111]. With (high) shear stress, cells show higher mitochondrial activity and increased expression of CYP450 enzymes, which highly impacts drug absorption and metabolism studies [27,112]. In GOCs, shear stress is applied to the cells by (1) pumping the medium through the microchannels using a (peristaltic) pump resulting in unidirectional flow or (2) using gravity driven flow resulting in bidirectional flow. From different studies [112,113,114] comparing the two different flow techniques or assessing multiple flow rates, it can be concluded that pump-driven unidirectional flow and high but physiological flow rates (~0.02 dyn/cm^2^) improve differentiation, barrier formation, mucus production and metabolic activity and thus might have the preference when selecting a GOC for your research.

When creating flow with a peristaltic pump, the peristaltic movement of the gut content is mimicked, which can be further stimulated by using fluctuating flow rates. The mechanistic stretch of the gut tubule itself is however only controlled in a couple of GOC designs by rhythmically applying computer-controlled vacuum suction to the chamber lateral to the microchannels [27,94]. Although these movement enhance villi formation, differentiation and increased barrier function, only a few GOC models incorporate peristaltic movements [27,74,75,76,77,93,106], probably (partially) explained by the need for a flexible chip material such as PDMS for this application.

Flexibility of the ECM, also described as matrix elasticity, is needed for (immune) cells to migrate through the ECM [115]. Therefore, matrix elasticity is particularly an important factor in GOC set ups that study inflammatory processes in the gut. Furthermore, increased matrix crosslinking and the release of metalloproteinases (MMPs) by the ECM negatively impacts the intestinal barrier function [115,116]. Although the matrix elasticity can be largely influenced by the choice of hydrogel(s) and crosslinking enzymes such as transglutaminase, the impact of different matrix elasticities on GOC performance has not been studied yet.

#### 2.4.2. Measurement of the Intestinal Barrier Function

With all the different mechanical cues impacting the barrier function of the intestinal epithelial layer, it is important to consider how this major function of the intestine can be assessed in the GOC. Furthermore, as good barrier integrity improves the credibility of GOC study outcomes, e.g., for the prediction of the oral bioavailability of drugs and treatment effects against leaky gut diseases such as IBD, measuring the barrier integrity is indispensable for GOC devices. For the GOC design, it is necessary to consider how this barrier function needs to be assessed. Basically, there are two ways to measure the barrier function: (1) assessment of the transepithelial electrical resistance (TEER) or (2) the use of (labeled) tracer molecules that assess paracellular and/or transcellular permeability.

TEER is easily assessed in Transwell studies using external electrodes; however, in GOC devices, this is technically more challenging as, in general, the apical and basolateral compartments are not easily accessible [117]. Therefore, electrodes might need to be integrated in the chip itself, which leads to many different techniques for all the different GOCs [27,96,104,118,119,120]. As a result, it is hard to compare different studies based on their reported TEER values. Furthermore, TEER is largely influenced by the choice for using ex vivo tissue (TEER of 50–100 Ω cm^2^) or cells (100–4000 Ω cm^2^) and also by the cell type, presence of a mucus layer and/or microbial biofilm [117,121].

The lack of standardization is also a problem when using tracer molecules to assess the intestinal barrier integrity. Different fluorescently, radiolabeled or unlabeled tracer molecules are employed, introducing a lot of variability with their variety in molecular weight (MW). Usually, one or more of the tracer molecules is applied apically and its/their permeability to the basolateral side is measured by taking samples from this compartment over time. Examples of commonly used fluorescently labeled tracers in GOC are rhodamine 123 (MW 380.8 g/mol) [24] and fluorescein isothiocyanate (FITC)-dextran (MW 4000–70.000 Dalton) [25,27,33,104]. Atenolol, mannitol, antipyrine and caffeine are examples of radiolabeled small molecule tracer compounds that can be used to assess the paracellular (atenolol, mannitol) or transcellular (antipyrine, caffeine) permeability and when different radiolabels are used they can even be paired to define the transcellular/paracellular transport ratio [17,25]. Whether the choice falls on TEER or tracer molecules to assess barrier integrity, the material and design of the GOC need to fit either the incorporation of electrodes or be non-absorbent for small molecules or a fluorescent dye, respectively.

#### 2.4.3. Control and Regulation of Physiochemical Parameters Oxygen and pH

Another group of parameters that affect the chip design are the physiochemical properties of the GOC environment. Relevant factors include oxygen, pH, temperature, CO_2_, and osmolarity. In GOC studies, the regulation and measurement of oxygen and pH levels receive the most attention. Tight control of the gastrointestinal pH, ranging from 5 to 7 in the various regions of the small and large intestine, is extremely important as biological processes are very sensitive to small deviations in the acid–base balance. Monitoring the pH in GOC systems is not incorporated in most designs. However, there are several possibilities with sensors or dyes. For example, metal-oxide base electrodes [122] or gold interdigitated electrodes [123] have been optimized for their compatibility with biological materials and are adapted to micro- or nanoscale technologies. Another method is the use of pH indicator dyes in combination with 2-photon microscopy, as demonstrated for mouse gastric fundus organoids with SNARF-5F [124] and for human duodenal organoids and colorectal cancer biopsies with SNARF-4F [125,126]. In vivo, the gut tissue receives only oxygen via its basolateral side whereas the gut lumen is almost completely anaerobic, thereby providing a suitable environment for the gut microbiome. Most in vitro GOC systems are not designed with an aerobic–anaerobic interface and thus do not control or monitor oxygen levels, but there are a couple of devices that incorporated oxygen sensors to monitor the oxygen gradient in the GOC. For example, the GOCs HMI, HuMiX, AOI Chip, anaerobic Intestine Chip, and GuMI integrated optical oxygen sensors into their device and show anaerobic conditions on the apical side of the cultured intestinal cells [78,87,103,104,105].

While oxygen concentrations are of vital importance for the survival of the intestinal and microbial cell populations in the GOC, pH changes can unintendedly be detrimental for the chip or hydrogel materials. In particular, chemically or physically crosslinked hydrogels are susceptible to pH changes. The ionic groups in the polymer backbone make polymers such as acrylic acid and methacrylic acid sensitive to protonation and deprotonation processes upon pH changes [127]. Under basic conditions, the carboxylic groups of acrylic acid and methacrylic acid are ionized, generating a repulsive force between the negatively charged surrounding groups, eventually leading to polymer swelling. Conversely, under acidic conditions, deionization of the same groups causes the network to collapse. Other hydrogels, such as poly(ortho ester) and poly(β-amino ester), contain moieties in their backbone that are cleaved upon a certain pH, making the polymers pH degradable [128]. Hydrogels that are pH-responsive are not necessarily unfavorable as this makes them more biocompatible, non-toxic, and biodegradable [129], but their applicability in GOC is rather limited by the narrow physiological range pH should have to maintain a healthy gut environment.

#### 2.4.4. Biochemical Cues

Soluble factors such as chemokines, cytokines, hormones, or other products of metabolic processes function as signal molecules in vivo. They are essential for cell survival, migration, proliferation and differentiation [130,131], but at this moment, only little attention is paid to incorporating such cues into GOC models. If applied, the administration of for example short chain fatty acids (SCFA) or cytokines is mainly used to modify the intestinal barrier integrity or to induce inflammation, frequently in the context of IBD. Examples of cytokine cocktails that have been used to induce inflammation and a so-called ‘leaky’ gut are TNF-α with IL-1β [95], TNF-α with IL-1β, IL-6 and IL-8 [89] or TNF- α with IL-1β and IFN-γ [132]. The effects of SCFA in GOC were studied in [78,133]. Whereas the first study provided the SCFA as pure compounds, in the latter study, butyrate-producing bacterial strain *F. prausnitzii* was co-cultured in the GOC to provide SCFA to the intestinal cells. Although both studies show anti-inflammatory effects of SCFA supplementation, contrasting results were found in the presence of activated immune cells [133]. This highlights the importance of choosing a proper and complete GOC design, for example also including biochemical cues, in order to provide a clear and translatable answer to the biological question.

Apart from providing biochemical cues to the GOC, the intestinal cells themselves are also producers of many soluble factors providing messages on for example their viability, inflammatory or metabolic state [131]. As messengers of the biological state of the GOC, these factors are called biomarkers. In order to measure biomarkers in the apical and basolateral fluids in a GOC system, the design should take into account that samples should be taken from both compartments and preferable at multiple timepoints in order to monitor changes over time. For most GOCs, sampling occurs manually through sample ports, medium flowing out of the GOC or accessing the apical and basolateral compartments directly. GOCs with one closed compartment, such as in a hollow fiber membrane design [113], or designs with bidirectional gravity-driven flow, such as the Organoplate^®^ [83,110], might not be suitable for many studies as they lack a sampling option from one of the compartment or show ambiguous results by mixing in- and outgoing fluids, respectively. The manual sampling can be labor-intensive and might introduce disturbing factors like temperature changes or air bubbles in the flow system when systems are relocated, or ports are opened for sampling. Automated sampling systems might offer a solution, but at this moment, such automated systems are not frequently utilized in GOC, or in any other OOC for that matter. There are, however, a few examples such as the NutriChip, that has a separate detection chamber dedicated to immunomagnetic assays using magnetic beads and fluorescent detection [134], and the coupling of the GOC with an online MS detection system [135,136,137]. Interestingly, these GOCs coupled to online chip-based MS detection systems were especially designed to assess the oral bioavailability of drugs such as curcumin [136], ergotamine and verapamil [137], and in combination with a digestion-on-a-chip model verapamil and omeprazole [135]. Surface-enhanced Raman spectroscopy (SERS) is suggested as another label-free optical-based read-out technique for biological molecules and might be implemented in future GOC models [138,139]. Although such automated sampling systems offer many advantages during the experiment, they also complicate the already complex design and setup of the GOC even further by the need for extra compartments, additional tubing, and connections to computers and other electronic devices. In the near future, most researchers will therefore probably remain using simpler GOCs with manual sampling.

## 3. GOC for Preclinical Drug Development Research

Having gone through all the different choices to obtain the best fit-to-your-needs GOC, it can then be used to answer the biological research question. Roughly, preclinical research using GOCs can be divided in (1) ADME studies using the GOC to study the fate of a drug, or (2) treatment-oriented research for intestinal diseases. In some cases, similar GOCs are used to find answers for both type of questions, but in general, GOC designs are substantially different with regards to chip materials and cellular components. Here, we include several examples of GOCs used in the context of oral drug delivery and in the area of gut health and disease.

### 3.1. GOC for ADME Research

For most drugs, the main route of application in patients is the oral route. The fraction that subsequently reaches the systemic circulation is defined as the oral bioavailability [140]. Bioaccessibility, which is the amount of digested compound available for absorption, intestinal absorption, and metabolism by gut and liver enzymes, includes the inevitable elements that contribute to the oral bioavailability of a drug candidate and is thus important to address in in vitro ADME research [141]. With the most common design of an intestinal layer of cells or tissue between an apical and basolateral compartment, most GOCs contain the elements to study intestinal absorption and gut wall metabolism.

The first proof of principal results on the applicability of a GOC for intestinal ADME research are usually on intestinal absorption only [25,137,142]. Although many GOCs assess the permeability of FTIC or FITC-dextran as a marker of intestinal integrity [25,27,33,86,93,95,96,142,143], only a few assess the intestinal absorption of therapeutic drugs. Examples are the assessment of verapamil and ergotamine epimers in a chip coupled to liquid mass spectrometry [137] and anti-cancer drug 5-FU in a chip containing tumor target cells in the bottom chamber [142]. As indicated, most of these studies limit their experiments to one or two drugs and do not provide data on the predictability of their model to accurately rank drug permeability from low to high, an important aspect for in vitro to in vivo translation. Using a GOC with ex vivo porcine or human intestinal tissue explants, Eslami Amirabadi and Donkers et al. showed a proper rank order relationship to known in vivo fraction absorbed values for 6 out of 7 tested drugs [25].

The added value of microfluidics on the metabolic capacity of cytochrome P450 3A4 was demonstrated in the Caco-2 Intestine Chip [144] and the Duodenum Intestine Chip [76]. With microfluidics, the Caco-2 cells presented increased CYP3A4 activity compared to Caco-2 cells on Transwell inserts [144], but only in the Duodenum Intestine Chip that used patient-derived organoid cells, CYP3A4 gene expression and protein level reached comparable levels to the adult human duodenum [86]. Although the latter system showed CYP3A4 gut wall metabolism by testosterone to 6β-hydroxytestosterone conversion, gut wall metabolism effects on the oral bioavailability of testosterone were not assessed. Remarkably and to the best of our knowledge, CYP3A4 seems to be the only metabolic enzyme so far for which the activity has been assessed in a GOC.

The bioaccesible aspect of oral drug bioavailability is usually neglected in GOC ADME research, as other in vitro models are usually used for this type of research [145]. Therefore, most studies provide the drug candidates in pure form and not in their therapeutic formulation. However, the impact of digestive processes on the chemical structure and breakdown of a drug molecule can be substantial, such as for peptide drugs [145]. Two early studies that included digestion in their setup designed a multi-OOC with additional compartments for liver cells and other organs [44,146], but did not specifically look at the effect of digestion on oral bioavailability. As a next step after intestinal uptake, digestion increased liver acetaminophen metabolism [44] and decreased the anticancer activity of tegafur, but not of cyclophosphamide, two anticancer prodrugs that need liver metabolism for their activation [146]. A more recent GOC connected a digestive compartment to an intestinal flow-through Transwell and did study the digestive effect on the bioaccessibility and intestinal absorption of omeprazole and verapamil [135]. They showed the digestive breakdown of omeprazole under acidic circumstances and thereby no availability of this drug for uptake in the intestinal compartment, whereas verapamil uptake was unaffected by the digestive processes [135].

Another oral bioavailability aspect usually absent in GOC designs is the impact of liver metabolism on the fate of a drug. In order to address this, the GOC needs to incorporate an additional chamber with liver cells or coupled to a separate liver-on-a-chip. Interestingly, many gut-liver-on-a-chips or multi-OOC including both gut and liver compartments have been described. Examples of such devices that focus on ADME research are the Physiomimix gut-liver-OoC that assessed the combined and individual contributions of the gut and liver in mycophenolate mofetil metabolism and used the in vitro data for in silico pharmacokinetic modeling [8], the Two-Organ-Chip platform (2-OC) describing acetaminophen absorption and metabolism [147], a microfluidic gut-liver chip demonstrating an improved representation of the flavonoid apigenin metabolic profile than for gut-only [148], coupled gut- and liver-OoCs that looked at diclofenac and hydrocortisone kinetics [149], a small intestine-liver-lung combination that evaluated the anti-cancer effects of two orally and one systemically administered drug [150] and a four-organ system consisting of intestine, liver, cancer and connective tissue models that demonstrated intestinal absorption, hepatic metabolism and cancer cell growth inhibition for 5-FU and 5-FU prodrugs CAP and tegafur [143]. Although most of these multi-OOCs have a focus on cancer therapy, one multi-OOC has been developed specifically for ADME studies and includes the four main organs in ADME: gut, liver, kidney and skin [151]. Sustained viability and multi-organ homeostasis was demonstrated for 28 days, which is substantially longer than the standard 1–3-day culturing period [151]. So far, no drug has been tested in this setting. Nonetheless, it is expected that this or other gut-liver-kidney-OoC combinations can substantially contribute to ADME profiling and pharmacokinetic modeling of novel drug entities.

### 3.2. GOC to Study Gut Health and Disease

Absorption and metabolism processes also play a role in the immunomodulatory effects of nutrients, as exposure to allergenic foods could trigger pathological symptoms, including gastrointestinal disorders, airway inflammation and red skin rashes [152]. To the best of our knowledge, no GOCs have been described to investigate allergen transport or allergen-induced epithelial cell activation. Nevertheless, expectations are high of a still-to-be-developed celiac disease-on-chip to improve disease development and design new therapies, especially because genetic differences among patients can be taken into account upon implementation of patient-derived iPSCs or biopsies in the GOC [153]. Interestingly, some nutrients can also be anti-inflammatory. Indeed, several food products, including orange juice, tomato and some dairy products reduce postprandial inflammatory responses [134,154]. The NutriChip was designed for the functional screening of foods with a focus on postprandial inflammation [154]. In this GOC, Caco-2 cells were co-cultured with a monocyte cell line (U937) differentiated into macrophages and stimulated with the endotoxin LPS. By measuring the secretion of pro-inflammatory cytokines IL-6 and IL-1β using a built-in immunomagnetic assay, a 1000-fold difference in LPS concentration was demonstrated to elicit a pro-inflammatory response by the macrophages [134]. Although intended to measure the anti-inflammatory effects of dairy products, this has not been demonstrated so far.

The major immune-related gastro-intestinal disease that has been studied in GOC is IBD. IBD is a collective term for a group of inflammatory diseases of the gut, including ulcerative colitis (UC) and Crohn’s disease (CD) [155]. It is believed that IBD is caused by a multitude of factors such as diet, stress, smoking, impaired bowel movement, genetic susceptibility, leaky epithelial barrier, with dysfunctional immune response to commensal microbiota as a leading cause. However, the exact disease mechanism is still largely unknown [155]. As the main characteristic of IBD is a leaky epithelial barrier, many GOC models first start to create this in their healthy (Caco-2) cell-based model using pro-inflammatory cytokines [89,95,132], bacterial endotoxins such as LPS [107,109,156] or chemicals such as dextran sodium sulfate (DSS) [96,157]. Pre-treatment of the GOC with anti-inflammatory compound TPCA-1 prevented cytokine-induced loss of intestinal barrier integrity and release of pro-inflammatory cytokines [95,156,158]. Likewise, pre-treatment with VSL#3 probiotics protected the GOC culture to DSS-induced colitis [96]. Remarkably, probiotic therapy was not successful if the IBD phenotype was already present and even ameliorated the pro-inflammatory response [96]. By having the possibility of individually and sequentially mimicking the different disease components (leaky epithelial barrier, immune cells, microbiota), the authors could break down the inflammatory response and assign certain processes to specific cell types, thereby showing the benefit of a GOC-based approach. Similar counterintuitive results were obtained for SCFA treatment in a gut-liver-immune chip that showed amplified inflammation and tissue damage in the presence of disease [133]. Positive treatment effects after disease onset were obtained for chitosan oligosaccharide and 5-ASA in a DSS- or *E. coli*-induced IBD-on-chip [157]. These studies show how difficult it can be to find an appropriate treatment for the different stages in such a complex disease, not even considering patient-to-patient differences. Fortunately, the GOC also offers the possibility to culture biopsy-derived organoids or iPSCs from healthy donors [158] or IBD patients [106,133] to recreate the IBD-phenotype on-chip, which was successfully demonstrated by retained disease characteristics on morphology and epithelial cell marker expression. As expected for patient-specific celiac disease-on-chip, patient-specific IBD-on-chip will support personalized treatment approaches.

In addition to the explorative research towards gut health effects of probiotics in (IBD-) GOCs, studies towards the therapeutic effect of synthetic biotics can be performed in a GOC as well. A synthetic biotic is a bacterium that has been genetically modified to perform a specific function for diagnostic or therapeutic purposes [159]. A synthetic biotic that already has advanced into clinical phase 2 studies is *E. coli* SYNB1618 for the treatment of phenylketonuria (PKU) [159]. PKU is a rare genetic disease caused by phenylalanine (Phe) hydroxylase (PAH) deficiency. If not controlled by strict diet-restrictions, PKU will lead to Phe accumulation in the blood and cause mental retardation, epilepsy and behavioral problems. Whereas the activity of one Phe inactivating enzyme of SYNB1618 is oxygen dependent, SYN5183 only expresses phenylalanine ammonia lyase (PAL) that converts Phe to non-toxic *trans*-cinnamic acid (TCA) in an oxygen-independent manner [160]. Using a GOC, a dose-dependent and dynamic conversion of Phe to TCA was shown, both when Phe was administered as oral bolus and when administered systemically, showing the enterorecirculation of Phe [160]. GOC data of the Phe flux in the presence of SYN5183 were subsequently used in computational models to describe Phe kinetics with SYNB1618 in vivo, demonstrating the added value of GOC research in therapeutic synthetic biotic development.

Other gastrointestinal or systemic diseases that were studied in GOC setting are enterititis caused by pathogen invasion [85], neonatal necrotizing enterocolitis (NEC-on-a-chip) [84], and following the recent pandemic outbreak SARS-CoV-2 (COVID-19) viral infection [91,161]. For each of these diseases, complex, and in some cases, new GOCs were designed that were tailored with a specific column [85], patient-derived cells [84,161] and microbiome [84]. Whereas the NEC-on-a-chip and one of the SARS-CoV-2 models only demonstrated the characteristics of these GOCs and predict their usability for treatment screening [84,91], antibiotic and probiotic therapeutic effects were evaluated in the ESBL-EC enteritis GOC [85] and six common antiviral drugs were tested against infection by SARS-CoV-2-related coronavirus NL63 [161]. Results in both studies were comparable to historical and clinical data, thereby highlighting the high translatability of GOC models to study gut health and disease.

Lastly, the mouth is also part of the GI tract and as a first contact with food and orally ingested drugs, similar to the intestines important for food digestion, drug absorption and GI health. Several oral mucosa-on-a-chip models have been described, usually consisting of keratinocytes, fibroblasts and sometimes endothelial cells to represent the gingiva in vitro [162,163,164,165,166]. Furthermore, challenge models have been developed to mimic typical diseases of the oral cavity such as oral mucositis [162] and periodontitis [165]. To improve the study of the latter disease, in which chronic inflammation of the gingiva is caused by a bacterial dysbiosis, co-culture models with oral bacteria have been established [163,166]. Additionally, the interaction the oral mucosa with dental materials and drugs is a major area of interest for the oral cavity [163,165]. Mechanical stimulation via fluid flow [164] as well as the possibility to increase culture time [162] are major advantages in these oral mucosa-on-a-chip models compared to their conventional predecessors, similar to the advantages of GOC models representing the intestines.

## 4. Discussion

With the establishment of so many different GOC models, of which as complete an overview as possible was given in this review, both the need for such complicated models for intestinal research, and at the same time, the broad applicability of the GOC in healthy and diseased settings are needed. The rapid developments and emerging number of GOC publications over the past decade show the wide interest and confidence in the improved tissue physiology and microenvironment in GOC compared to conventional in vitro gut models. Furthermore, the potential to include patient-derived intestinal cells or tissue explants as well as other patient-derived components such as immune cells and microbiome, provide an adjustable platform for personalized medicine approaches. Nevertheless, the almost unlimited choices in combinations of chip component materials, cell types and sources, environmental and biochemical cues, and (integrated) read out possibilities prevent standardization and hamper the adoption of GOC models in the industrial preclinical drug development pipeline by pharmaceutical companies [167,168,169,170]. As the intended end-users of the GOC for oral bioavailability studies and therapeutic drug design, pharmaceutical industry expects acceleration in the lead optimization process and budget savings of 10–26% in research and development costs upon OOC implementation [171]. This is not a small feat considering the $1 billion per new drug entity reaching the market [168].

Whereas most GOC and OOC models are used for in-house research only, currently around 20–25 devices have become commercially available. A recent survey indicated that the GOC models developed by Mimetas (Organoplate^®^) and Emulate (for example Intestine Chip) have been used by a few big pharma companies [169]. Nevertheless, the contribution of OOC research to ongoing candidate drug development remains elusive and adaptation of these models progresses slowly [169]. The major restraints are (1) costs, (2) ongoing model developments, (3) standardization, and (4) regulatory acceptation. Compared to conventional in vitro gut models, GOC experiments are more expensive because they require more cells and reagents per entity, have a lower throughput and are labor-intensive in system setup and sampling. Furthermore, companies need to do large financial investments to obtain the technology in-house and train their employees without knowing if the invested platform will turn out to be the leading GOC in the future [167]. At this moment, there are just too many possibilities and with the ongoing development, many of the GOCs might not even have reached their final design. Additionally, the level of characterization for GOC models is far behind the level of characterization of conventional in vitro gut models. For example, intestinal absorption has been assessed for >100 compounds in Caco-2 Transwell studies, whereas only a handful of compounds have been tested in the varies GOCs. Furthermore, the aspects that are characterized differ from chip to chip, making it hard to compare the different setups, especially since the chip material and cells used vary a lot. This lack of standardization, limited use of scalable technology, and high costs per device, not only hampers GOC characterization, but also discourages regulatory agencies to include OOC research as a milestone in regulatory submissions of drugs [167,170].

The ultimate adoption of one, two or a couple of GOC designs, depending on the research question, as a novel golden standard research tool in future intestinal research will likely take many years. In order to accelerate GOC standardization and implementation in pharmaceutical research, a systematic and collaborative approach with representatives from academia, industry and government would be recommendable. By working closely together in an open environment, knowledge gaps can be closed faster, and GOC characterization will be ameliorated. First attempts of such open technology programs with multiple stakeholders are promising, for example the Translational Organ-on-Chip Platform (TOP) that aims to provide an infrastructure for OOC plug-and-play [172] or the Organ-On-Chip project within Moore4Medical that develops a ‘smart multiwell plate’ [173]. Nevertheless, it should not be forgotten that the currently existing GOCs already hold a great potential to perform intestinal ADME or gut health and disease studies. As such, the GOC will be a leading tool shaping future intestinal in vitro research and contribute largely to novel drug development.

## Figures and Tables

**Figure 1 biomimetics-08-00226-f001:**
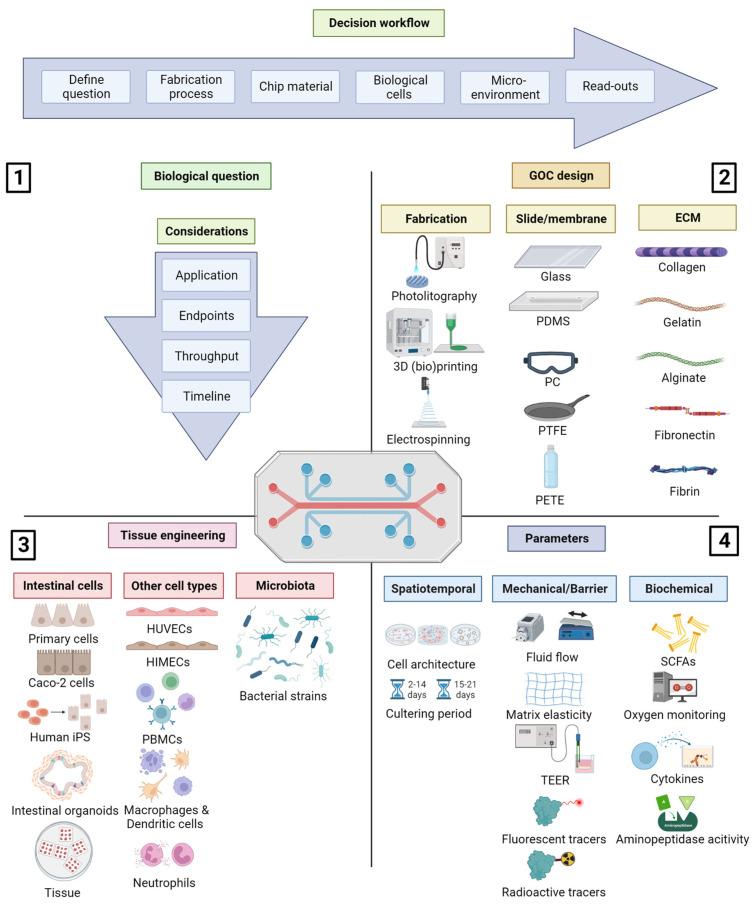
Overview of the four most important components to consider when establishing a gut-on-a-chip (GOC). The decision workflow is depicted above the four quadrants. (**1**) The biological question is important for the characteristics the GOC must comply to, determining the choices to make in quadrants 2, 3 and 4. (**2**) Fabrication and materials used for the chip slide, membrane and extracellular matrix (ECM). (**3**) Tissue engineering entails the different cell types that are cultured on the GOC. (**4**) Environmental factors and read-out parameters. Figure created with BioRender.com, accessed on 19 April 2023.

**Figure 2 biomimetics-08-00226-f002:**
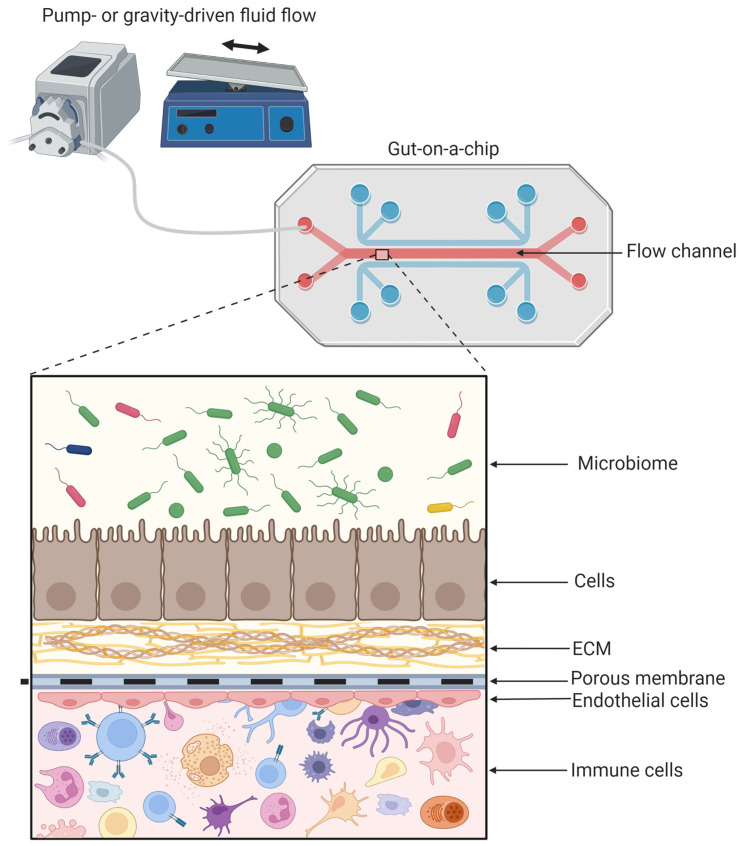
Common components of a gut-on-a-chip (GOC). In the flow channel, enterocytes grow in the apical chamber, separated from the endothelial cells by a porous membrane coated with extracellular matrix (ECM). Optionally, microbiota or immune cells can be added to the apical or basolateral channels, respectively, to add more complexity to the model. Figure created with BioRender.com, accessed on 19 April 2023.

## Data Availability

No new data were created or analyzed in this study. Data sharing is not applicable to this article.

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
