# Peer review of "Gut-on-a-Chip Research for Drug Development: Implications of Chip Design on Preclinical Oral Bioavailability or Intestinal Disease Studies"

_biomimetics, 2023, doi:10.3390/biomimetics8020226_

Round 1

Reviewer 1 Report

Recently, researchers have preferred ChIP models over conventional models to study visceral organ cross-talk with other organs. However, interactions among different cell types and their structures remain a challenge. In this review article, the authors give step-by-step options for selecting Gut-on-a-Chip (GOC) to work on various research questions. Although the manuscript is well written, some concerns are mentioned here that need to be addressed.

1.    While studying the cross-talk of the intestinal epithelium with other cell types that have different food requirements, it is not described how to supply them simultaneously without affecting the other cell type.

2.    While studying specific microbial effects on the integrity of the gut, antibiotics are not used at the time otherwise they would destroy them. In this case, there may be a possibility of mycoplasma and other bacterial contamination while doing cell culture on the chip. The author should discuss this issue as well.

3.    Authors have discussed about the hypoxic chip model for an anaerobic microbial-gut experiment. However, in the case of the gut-heart axis how to assemble the gut-cardiomyocyte-on-ChIP to study the effect of microbial flora on gut-heart co-culture if we do not wish to make cardiomyocytes hypoxic? Including this can be very helpful for the readers.

4.    Label-free optical-based read-out techniques, such as surface-enhanced Raman spectroscopy (SERS), need to be discussed in order to make accurate and real-time decisions while overcoming the manufacturing drawbacks of GOC models (PMID: 36671971).

5.    Authors have not discussed the pH sensors being used recently in Gut-on-a-ChIP, such as two-photon microscopy with SNARF4F dye, etc.

6.    In the introduction section, the authors have mentioned that the advantages and disadvantages of the four components influencing GOC design will be discussed. Including a table for all components with their advantages and disadvantages will be of a wide readership. 

7.    Several abbreviations are not clear in line number 412-413. These should be written as PMI ChiP (physiodynamic mucosal interface-on-a-chip), HMI (human-microbial interface), HuMiX (human-microbial crosstalk), etc.

It is OK and understandable for the general audience.

Reviewer 2 Report

Overall, the manuscript is well written is excellent schematics and organization. 

Minor revision: While the authors have presented literature related to the gut in general with most of the literature focussed on small/large intestinal models. The gastrointestinal system per se includes the oral cavity/mouth, where the digestion of food and rapid absorption of small molecules such as glucose starts. Hence authors to consider including a discussion on the use of microfluidic systems to study host-material and host-microbe interactions using oral mucosa/gingival models. Some of the key literature include - DOI: 10.1088/1758-5090/ac933b, 10.1063/1.5048938, 10.1002/adhm.202202376, 10.3390/bios12050345, 10.3390/bioengineering10050517
